# Seasonal Variability of Nutrients and Radium Isotope Fluxes from Submarine Karstic Spring at the Southwest of Crimea, Black Sea

Illarion I. Dovhyi [1,*], Ol'ga N. Kozlovskaia [1], Nikolay A. Bezhin [1], Iuliia G. Shibetskaia [1], Aleksey I. Chepyzhenko [1] and Ivan G. Tananaev [2,3]

1   Marine Hydrophysical Institute, Russian Academy of Sciences, Kapitanskaya Str., 2, 299011 Sevastopol, Russia; o.n.kozlovska@gmail.com (O.N.K.); nickbezhin@yandex.ru (N.A.B.); iuliia.shibetskaia@gmail.com (I.G.S.); ecodevice@yandex.ru (A.I.C.)
2   Department of Chemistry and Chemical Engineering, Sevastopol State University, Universitetskaya Str., 33, 299053 Sevastopol, Russia; tananaev.ig@dvfu.ru
3   Department of Nuclear Technology, Far Eastern Federal University, Sukhanov Str., 8, 690091 Vladivostok, Russia
*   Correspondence: dovhyi.illarion@yandex.ru

**Abstract:** The groundwaters of the southwestern region of Crimea are formed on the karst plateaus of the Crimean Mountains, and a significant amount of them is discharged into the Black Sea. The Crimean Peninsula is a water-deficient region; therefore, the study of its hydrogeology is an urgent task, since groundwater is a valuable freshwater resource. Through submarine groundwater discharge (SGD), the transfer of chemical compounds to the sea in the event of anthropogenic interference may also occur. In this work the fluxes of submarine groundwaters in the area of Cape Peleketo in different seasons, and also the fluxes of nutrients within them, are evaluated for the first time, as well as the factors determining their variability. During the study, hydrological (temperature, salinity (S), current velocity) and hydrochemical (concentration of biogenic elements) parameters, as well as the concentration of long-lived isotopes of $^{226}$Ra and $^{228}$Ra, were measured. The SGD fluxes were estimated through the mixing formula. As the endmember, we used groundwater concentrations of nutrients or radiotracers, defined by extrapolation of nutrients or radium concentrations to zero salinity. Significant differences in the studied region's SGD flux values (from 4100 to 13,900 m$^3$/day) are shown; maximum values are in winter and summer, and minimum values are in autumn and spring. The relationship between the seasonal variability of the discharge intensity and the amount of precipitation in the groundwater formation area is shown. The data obtained show that this source makes a significant contribution to the local supply of nutrients. Substantial amounts of nitrates come from the karst cavity, which can lead to eutrophication and limit the primary production of phosphorus in the local coastal sea region.

**Keywords:** submarine discharge; seasonal variability; hydrological; hydrochemical parameters; nutrients; long-lived radium isotopes



## 1. Introduction

The study of SGD is one of the most important problems of modern hydrogeology and oceanology. Recent reviews devoted to this problem [1–4] indicate a large increase in the number of publications in 2000–2020. SGD makes a significant contribution to the migration of nutrients in coastal waters [1,5,6]. Numerous submarine springs are known in the Mediterranean region [7]. Some researchers consider that submarine springs are the main source of nutrients in the Mediterranean [8,9]. Submarine springs in the Black Sea are deficiently studied [7–10]. Some springs have been described on the southwestern coast of the Crimean Peninsula [11–13] and Romania [14], numerous submarine springs are known

on the coast of the Caucasus [10,14,15], and an extensive submarine depression is located off the coast of Abkhazia [15]; it is formed by the discharge of the Arabika Massif. However, the number of papers dedicated to submarine springs in the Black Sea is negligible in comparison with that for the Mediterranean.

Normally, complex studies are carried out to study SGD. Hydrological parameters (temperature and salinity), and hydrochemical parameters (concentration of nutrients, stable isotopes $\delta^{18}$O, $\delta^2$H [16], natural radionuclides—$^{222}$Rn [16,17], isotopes of Ra [3,18,19]) help assess the fluxes of SGD and associated solutes. Remote sensing methods have recently been widely used to search for new springs of SGD [20].

The area of southwestern Crimea includes the ending of the Crimean Mountains that arose during the neotectonic activation at the location of the Cretaceous-Paleogene denudation plain and the adjacent shallow-water carbonate sedimentation basin. The Crimean Mountains are one of the links of the Alpine-Himalayan orogenic belt formed during the collision of the Eurasian, African, and Indo-Australian plates.

The main ridge of the Crimean Mountains is composed of the Upper Jurassic deposits; in the coastal cliff, they are exposed to the west of Cape Ayia. The Upper Jurassic is a complex sedimentary complex of marine origin. Lithologically, the Upper Jurassic complex of rocks is represented by various types of carbonate, clayey, and terrigenous formations that are complexly interconnected with each other. The investigated region is located within the hydrogeological region of fractured karst waters of the Crimean Mountains. The geological structure and hydrogeology are described in detail in the work [21].

The object we investigated is a karst cavity located under a rocky cliff, near Cape Ayia on the southwestern coast of the Crimean Peninsula, which was previously studied in [11,12]. The information on the SGD fluxes from the karstic cavity was obtained based on the data on salinity and concentration of silicates.

Radiotracer methods have been used for a long time to search for the centers of SGD [3] and are recommended for studying SGD in coastal areas [18,19]. Short-lived ($^{223}$Ra, $^{224}$Ra) and long-lived ($^{226}$Ra, $^{228}$Ra) radium isotopes of terrigenous origin have become most widespread for assessing the flux of submarine springs. Conclusions about SGD as a source of nutrients in the Mediterranean region were made based on the study of the $^{228}$Ra balance, as well as based on the correlation between the concentration of radium isotopes and nutrients [9].

The spatial distribution of $^{226}$Ra in the Black Sea was studied in [22–24]; data on the distribution of $^{228}$Ra in the Black Sea are scarce [25,26], but the concentration of $^{228}$Ra in the Sea of Marmara was given in [9].

The Crimean Peninsula is a water-scarce region, which is why the study of the groundwater balance, its state, and interaction with seawater is an urgent task. From a practical point of view, it is useful to study the possibility of capturing submarine springs, since the use of submarine groundwater for economic needs has been shown to be economically feasible in some countries of the Mediterranean basin. Such technologies were developed by MarineTech [27].

This source has been suggested for a long time as applicable for capture. In the 1990s, an attempt was made to close the exit from the cave with a metal gate to prevent mixing of fresh- and seawater. However, the work was not completed, and the roughness of the sea carried away the structure.

Due to the lack of fresh water in Crimea, these works are relevant. Previously [28], we described the distribution and fluxes of nutrients and isotopes from the submarine source described in this article in the spring period of 2019. The aim of this work is to study the current state of the submarine spring near Cape Ayia, Sevastopol region (Figure 1), assess its fluxes in different seasons, identify the reasons for its variability, and assess the state of groundwater that forms this spring.

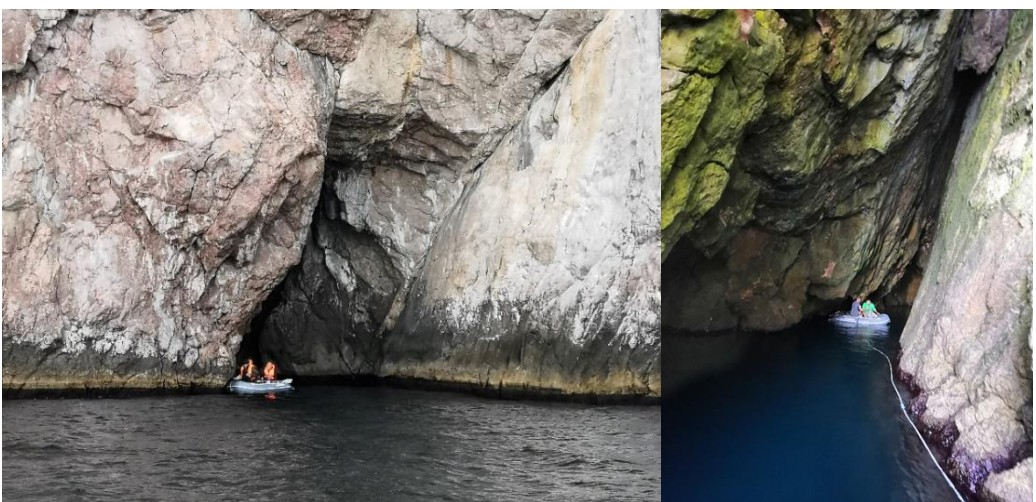

**Figure 1.** "Ekaterininsky grotto" near Cape Peleketo (research area near Cape Ayia). Photo taken by 24 March 2019.

## 2. Materials and Methods

### 2.1. Coastal Expedition and Sampling

Descriptions of hydrological measurements, concentrations of $^{226}$Ra and $^{228}$Ra from sea water, analysis of isotopes of $^{226}$Ra and $^{228}$Ra and biogenic elements—silicic acid, nitrates, nitrites, ammonium, DIP (dissolved inorganic phosphorus), and TDP (total dissolved phosphorus), are given in [28,29]. A certified soil source with known specific activity IAEA CU-2006-03 was used to calibrate the gamma spectrometer. The relative determination error was 1.5–2% for DIP and TDP (concentration range 0.2–8 µM), 4–15% for $NH_4^+$-ions (concentration range 0.2–1.0 µM), 0.13–2% for silicic acid (concentration range 1.1–18.8 µM), and 0.01–0.1% for nitrates and nitrites (concentration range 0–1 µM).

The dependences of the concentration of nutrients on salinity based on the literature data were obtained using the program GetData Graph Digitizer version 2.26.0.20.

Coastal expeditions to Cape Ayia took place on 24 March 2019, 10 September 2019, 23 February 2020, and 19 July 2020. Surface water samples were taken at the Ayazma-Chokrak spring on 29 September 2019 (coordinates N44.47079 E33.64401).

Hydrological measurements were carried out at 20–24 stations (Figure 2) in the karstic cavity near Cape Peleketo and the vicinity aquatory. Samples were taken to determine the concentration of nutrients at 24 stations.

Surface water samples for determining the concentration of nutrients were taken in 125 mL plastic containers. Within 6 h, the samples were delivered to the onshore laboratory for analysis without the addition of substances, suppressing the vital processes of microorganisms. The samples were filtered through membrane filters with a pore diameter of 0.45 µm (Vladisart), and analyzed on the same day.

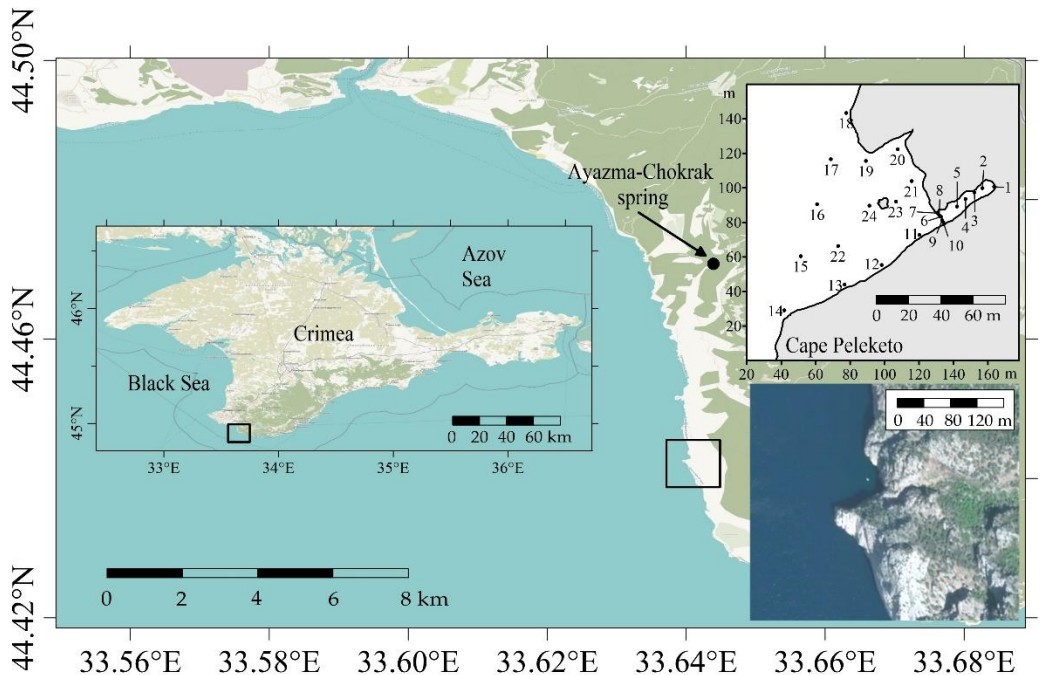

**Figure 2.** Location and layout of sampling stations.

*2.2. Calculations*

The submarine groundwater flux was calculated using a mixing formula using the concentration of radium isotopes, nutrients, or salinity as parameters.

$$Q = \int \int_{00}^{HL} U_{av}(x,y) \frac{A_{bg} - A(x,y)}{A_{bg} - A_n} dxdy, \tag{1}$$

where $Q$ is a flux of the submarine spring; $U_{av}(x,y)$ is the average current velocity at point $x, y$ of the section; $L, H$ is maximum width (7 m) and thickness of the surface layer of the brackish water (0.5 m); $A_{bg}$ is the background parameter value; $A(x,y)$ is the parameter value at the point $x, y$ of the section; $A_n$ is the average parameter value in freshwater. The vertical distribution of the current velocity at station 6 in other expeditions is presented in Table 1.

**Table 1.** The current velocity in the section of the grotto (station 6) at different depths.

| 23 March 2019 | | 10 September 2019 | | 23 February 2020 | | 19 July 2020 | |
|---|---|---|---|---|---|---|---|
| **H, m** | ***v*, cm/s** | **H, m** | ***v*, cm/s** | **H, m** | ***v*, cm/s** | **H, m** | ***v*, cm/s** |
| 0.2 | 11.9 | 0.2 | 6.0 | 0.1 | 11.0 | 0.1 | 17.0 |
| 0.5 | 12.2 | 1.2 | 5.5 | 1.4 | 7.0 | 1.1 | 9.0 |
| 4.0 | 11.6 | 3.3 | 5.5 | 3.7 | 7.0 | 3.5 | 13.0 |
| 7.7 | 12.0 | 6.7 | 3.5 | 6.1 | 3.0 | 7.1 | 2.0 |

Nutrient fluxes were calculated as follows:

$$F = Q\,C, \tag{2}$$

where F is the flux of an element, g/day; Q is the mean value of the flux of the submarine spring, L/day; C is the concentration of an element, g/L.

## 3. Results and Discussion

### 3.1. Ra Isotopes

Figure 3b,c show the distributions of the activity of the $^{226}$Ra, $^{228}$Ra isotopes in the samples taken during coastal expeditions.

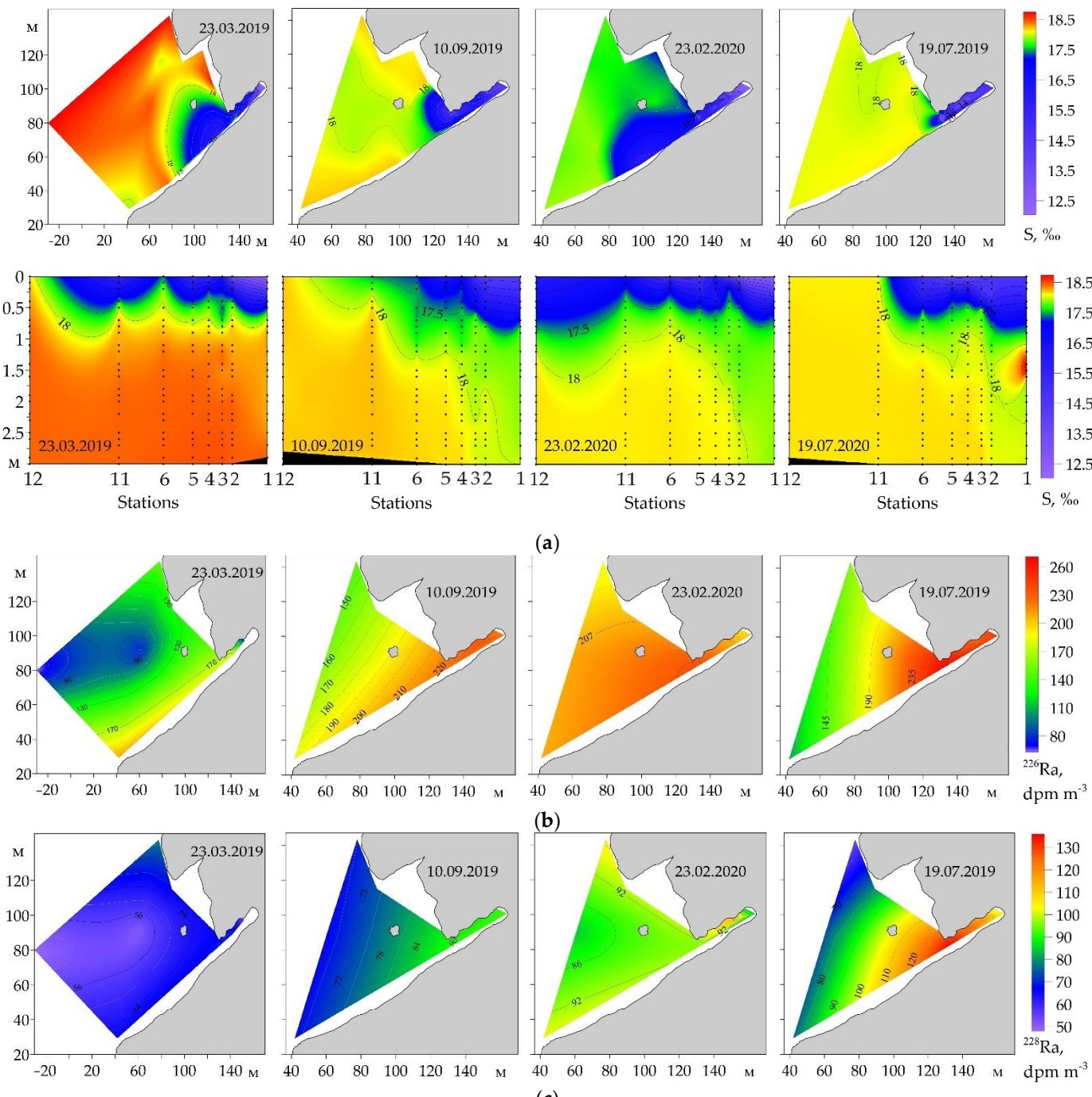

**Figure 3.** *Cont*.

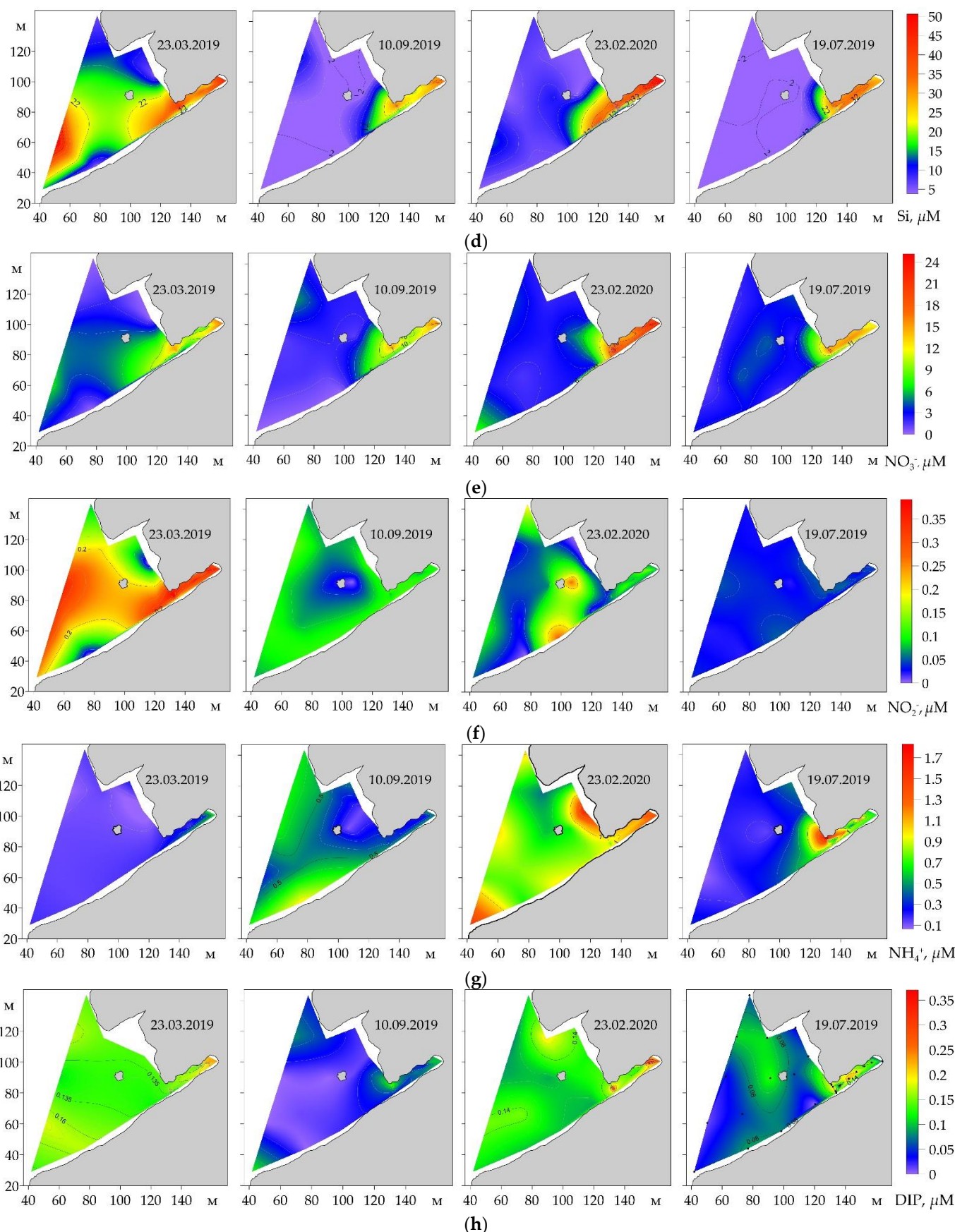

**Figure 3.** *Cont.*

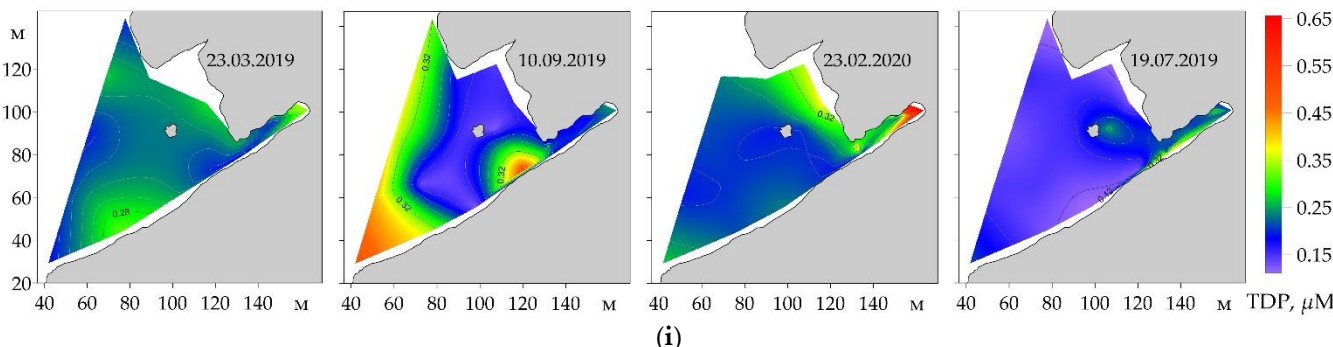

**(i)**

**Figure 3.** Salinity in the surface layer and at the section of stations 1–12 (**a**), concentrations of $^{226}$Ra (**b**), $^{228}$Ra (**c**), silicic acid (**d**), nitrates (**e**), nitrites (**f**), ammonium (**g**), DIP (**h**), and TDP (**i**) in the karst cavity and adjacent stations.

The object under study is a karst cavity open to the sea on one side. Therefore, earlier [11–13], researchers suggested using the mixing formula to determine the flux (1).

When calculating the flow of submarine groundwater in different seasons, the following were used: the concentration of radium isotopes at station 6, in groundwater, and background values for the region.

The background activity values of $^{228}$Ra or $^{226}$Ra are equal to $50 \pm 15$ dpm/m$^3$ (disintegration per minute) and $61 \pm 21$ dpm/m$^3$, respectively, according to the data of the 106 cruise of the RV Professor Vodyanitsky [26]. W.S. Moore [24] gives average values for the concentration of $^{226}$Ra in the surface layer of the Black Sea: $1.34 \pm 0.21$ Bq/m$^3$.

The activity values of the $^{228}$Ra or $^{226}$Ra in the groundwater were 189 and 394 dpm/m$^3$; these values were obtained by extrapolating the dependences of the concentration of radium isotopes on salinity according to the data of four expeditions (Figure 4). The concentration of $^{226}$Ra in the rivers of the Black Sea region is 0.42–3.9 Bq/m$^3$ [24]; however, in groundwater, the concentration of $^{226}$Ra is usually higher [30], due to the longer time of contact with rocks.

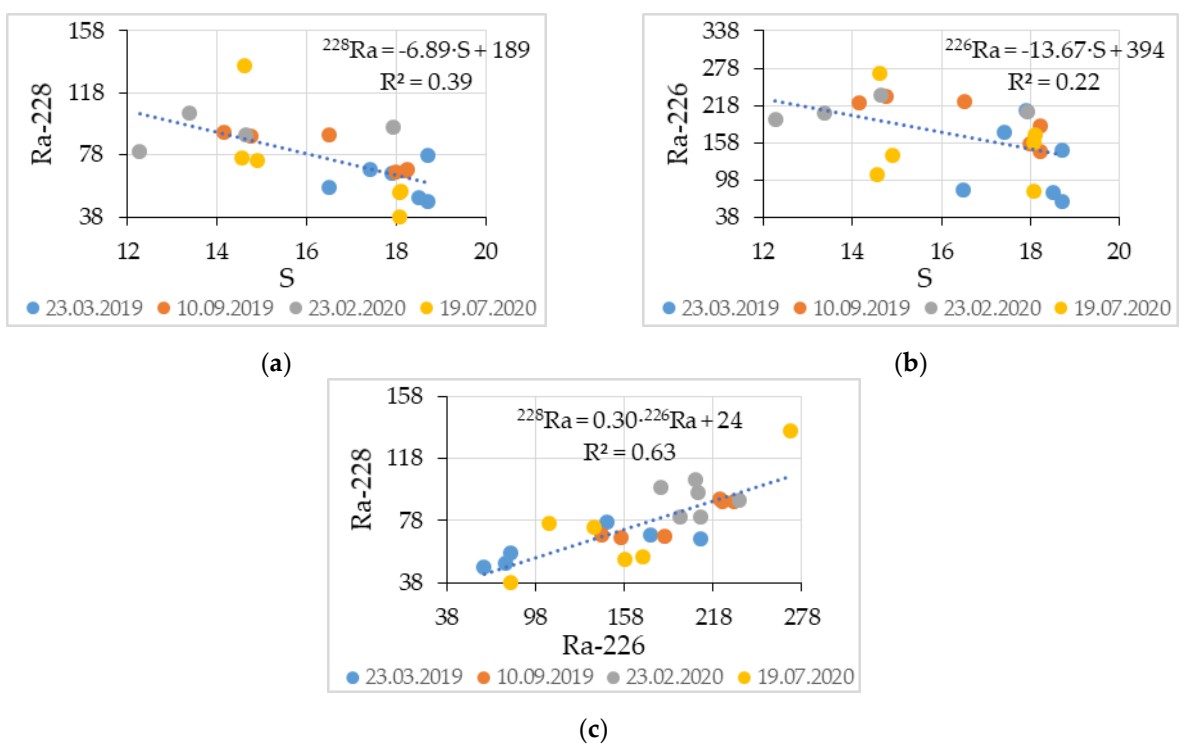

**Figure 4.** Dependence of the concentration of $^{228}$Ra (**a**) and $^{226}$Ra (**b**), and the ratio of isotopes $^{228}$Ra and $^{226}$Ra (**c**) according to the results of 4 expeditions.

The values of SGD fluxes in different periods calculated using the concentration of radium isotopes are shown in Table 2.

**Table 2.** Assessment of SGD and nutrient fluxes.

| Season | Submarine Water Fluxes Calculated by the Parameters, m³/day | | | | | | | | Nutrient Fluxes, g/day | | | | | |
|---|---|---|---|---|---|---|---|---|---|---|---|---|---|---|
| | $^{226}$Ra | $^{228}$Ra | S | Si | DIP | NH$_4^+$ | NO$_3^-$ | Mean | Si | DIP | TDP | NH$_4^+$ | NO$_2^-$ | NO$_3^-$ |
| Spring | 7530 | 7960 | 7235 | 9540 | 9930 | 5390 [2] | 7110 | 8220 | 23,980 | 97 | 115 | 135 | 106 | 5360 |
| Autumn | 4030 | 4170 | 4350 | 3735 | 6740 [2] | 4750 | 4050 | 4100 | 16,600 | 45 | − [1] | − [1] | 10 | 3650 |
| Winter | 7780 | 8390 | 7645 | 10,520 | 14,415 [2] | 8610 | 12,490 | 9360 | 35,300 | 170 | 330 | 270 | 29 | 7350 |
| Summer | 13,920 | 14,350 | 14,030 | 15,250 | 22,350 [2] | 33,330 [2] | 15,400 | 13,900 | 52,100 | 222 | 226 | 480 | − [1] | 9950 |

[1] It was impossible to calculate reliably since, due to the processes of microorganism activity, the concentration of nutrients is higher at the background stations than in the grotto. [2] Was not used to calculate the mean, since bias is assumed due to the activity of marine organisms.

Station 6 (at the section of the karst cavity) was chosen for calculating the fluxes due to the following reasons. According to the published data [11–13] and our observations, there are two outlets of groundwater in this karst cavity—in the innermost part and at the mouth of it (stations 1 and 10 on Figure 2). This explains the presence of two zones with brackish water and two zones with high concentrations of nutrients in the cavity. Salinity is higher at stations 3 and 4 between these zones, and the concentration of nutrients and radium at station 3 is lower than at station 6.

The flux value derived from salinity and silicic acid concentration is 1915 m³/day, obtained in September 2007 after an abnormally hot and dry summer [13].

The obtained values of $^{228}$Ra and $^{226}$Ra activity correlate poorly with salinity (Figure 4) because of the least accuracy among all the studied parameters.

Twenty-four samples for nutrients were collected. All the measured parameters in the salinity range of 12–19‰ were noticeably scattered. Increased values of some parameters were observed at stations remote from the karst cavity. This is explained by the removal of brackish waters at a considerable distance from the source.

The analysis of the $^{228}$Ra/$^{226}$Ra ratios is of great interest. It follows from the analysis of the published data [30] that the $^{228}$Ra/$^{226}$Ra ratio is higher in seawater, since the concentration of $^{228}$Ra in it is higher than that of $^{226}$Ra, and vice versa in freshwater. Thus, the $^{228}$Ra/$^{226}$Ra ratio equal to 0.9 was obtained for "seaward" samples obtained in the 106 RV Professor Vodyanitsky cruise, and 0.32 for the samples near the submarine spring (Figure 4c). Note that salinity in this part of the study area [30] was oceanic 33–36 ‰, which is higher than in the study region overall. Similarly, in this work, lower $^{228}$Ra/$^{226}$Ra values were observed at the background station 5 opposite the exit from the Balaklava Bay during the 106 RV Professor Vodyanitsky cruise. This is caused by a large discharge of fresh sewage water (about 3 million m³ per year) [31]. $^{228}$Ra/$^{226}$Ra values were also lower at station 1 and station 6 near the karst cavity, which is due to the large proportion of fresh water in the sample. Correspondingly, higher $^{228}$Ra/$^{226}$Ra ratios were observed for the samples with a larger proportion of seawater.

### 3.2. Nutrients

Dissolved silicic acid in groundwater comes from the weathering of surrounding rocks. Its typical concentration in groundwater is 15–350 μM [32] and mainly depends on the types of rocks or soil and is less variable than other nutrients.

In [11,12], the concentration of silicic acid was used as a tracer of SGD in the study area, and in [13] phosphate concentration was also used. The present data lead to the following relationships:

$$Si = -6.1 \cdot S + 113.5 \text{ (September 2007),} \tag{3}$$

$$Si = -6.76 \cdot S + 124.3 \text{ (August 1994),} \tag{4}$$

$$Si = -6.1 \cdot S + 109.6 \text{ (autumn 1993)}. \tag{5}$$

The equations obtained by us in different seasons for this region are close to those in the literature (Table 3).

**Table 3.** Seasonal variability of the dependences of nutrient concentrations in the surface layer on salinity and concentrations of nutrients in the "groundwater".

| Season | Dependence of the Concentration of Nutrients on Salinity and Squared Correlation Coefficient for Each Dependence | | | | | | Concentration in "Groundwater", μM | | | | | |
|---|---|---|---|---|---|---|---|---|---|---|---|---|
| | Si | DIP | TDP | $NH_4^+$ | $NO_2^-$ | $NO_3^-$ | Si | DIP | TDP | $NH_4^+$ | $NO_2^-$ | $NO_3^-$ |
| Spring | $107.0–5.1 \cdot S$ $R^2 = 0.57$ | $0.38–0.01 \cdot S$ $R^2 = 0.54$ | $0.46–0.01 \cdot S$ $R^2 = 0.25$ | $1.22–0.06 \cdot S$ $R^2 = 0.71$ | $0.95–0.04 \cdot S$ $R^2 = 0.61$ | $48.3–2.5 \cdot S$ $R^2 = 0.35$ | 103.5 | 0.37 | 0.45 | 1.18 | 0.92 | 46.6 |
| Autumn | $149.9–8.0 \cdot S$ $R^2 = 0.77$ | $0.36–0.02 \cdot S$ $R^2 = 0.3$ | $0.02 + 0.01 \cdot S$ $R^2 = 0.02$ | $-0.38 + 0.05 \cdot S$ $R^2 = 0.08$ | $0.19–0.01 \cdot S$ $R^2 = 0.76$ | $66.1–3.5 \cdot S$ $R^2 = 0.07$ | 144.3 | 0.35 | $–^1$ | $–^1$ | 0.18 | 63.6 |
| Winter | $162.0–8.5 \cdot S$ $R^2 = 0.74$ | $0.70–0.03 \cdot S$ $R^2 = 0.55$ | $1.35–0.06 \cdot S$ $R^2 = 0.72$ | $2.43–0.10 \cdot S$ $R^2 = 0.58$ | $0.27–0.01 \cdot S$ $R^2 = 0.60$ | $67.3–3.4 \cdot S$ $R^2 = 0.4$ | 156.0 | 0.68 | 1.31 | 2.36 | 0.26 | 64.9 |
| Summer | $115.9–6.1 \cdot S$ $R^2 = 0.61$ | $0.50–0.02 \cdot S$ $R^2 = 0.45$ | $0.47–0.02 \cdot S$ $R^2 = 0.45$ | $2.41–0.11 \cdot S$ $R^2 = 0.24$ | $0.002–0.002 \cdot S$ $R^2 = 0.74$ | $50.6–2.6 \cdot S$ $R^2 = 0.07$ | 111.6 | 0.49 | 0.46 | 2.33 | $–^1$ | 48.7 |

[1] It was impossible to calculate reliably since, due to the processes of microorganism activity, the concentration of nutrients is higher at the background stations than in the grotto.

For Geoje Bay [33], the equation is as follows:

$$Si = -6.8 \cdot S + 234 \tag{6}$$

It should also be noted that, although in many papers [6,34–37] graphs of the dependence of the concentration of nutrients and radiotracers on salinity are given, this relationship is not analyzed mathematically. Extrapolation of the dependence to the salinity of fresh water (0.7), which makes it possible to estimate the content of nutrients and radiotracers in it, is also not performed. We independently performed the processing of the graphs given in paper [6]. The dependence of silicic acid on salinity there is described by Equation (7), and is close to those obtained by us (Equation (8)) when processing our data from all four expeditions.

$$Si = -3.6 \cdot S + 137 \tag{7}$$

$$Si = -6.7 \cdot S + 131 \tag{8}$$

Aqueous inorganic phosphorus comes to groundwater by mineral leaching, decomposition of organic matter in soils, or anthropogenic activity.

The ratio of the concentration of dissolved inorganic phosphorus and salinity obtained in [13] is as follows:

$$DIP = -0.019 \cdot S + 0.349 \text{ (September 2007)}, \tag{9}$$

The equations obtained for each expedition are given in Table 3, and for all four:

$$DIP = -0.01 \cdot S + 0.28, \tag{10}$$

However, the value of the squared correlation coefficient for four seasons is only 10%, which is much lower than that calculated for each season. This is due to the difference in the concentration of phosphorus in the surface layer, due to its involvement in various biogeochemical processes including sorption reactions between phosphorus and the carbonate matrix of aquifers [38].

Nitrates are the most common groundwater pollutants, and the origin of their input can be both natural and anthropogenic factors.

Comparison of the data extracted from [6] (11) and summarized by us for all seasons (12) shows that the contribution of pollution of groundwater in the northern Mediterranean is higher than in the southwestern Crimea.

$$NO_3^- = -2.7 \cdot S + 103.8, \tag{11}$$

$$NO_3^- = -3.0 \cdot S + 58.3, \tag{12}$$

Similarly, the fluxes of submarine groundwater were calculated according to the mixing formula using salinity and concentration of nutrients (Table 2). The background values of DIP, silicic acid, ammonium, nitrite, and nitrate were taken as the lowest values in a given season. Extrapolation of nutrient concentrations to zero salinity gives the assumed concentrations of nutrients in groundwater (Table 3). The values of the daily water discharge in the submarine spring using ammonium, TDP, and nitrite ions differ greatly from the others (Table 2). At the same time, for TDP and nitrite ions, low values of the approximation reliability are observed. Note that these parameters—reduced forms of nitrogen and total dissolved phosphorus, which include dissolved organic phosphorus as a component—largely depend on the vital activity of various microorganisms. This is the reason for the poor correlations of these parameters with the salinity.

The concentrations of nutrients and the average flux of the submarine spring allow us to estimate the fluxes of nutrients.

The fluxes of nutrients were calculated according to Equation (2) for other periods; the data are given in Table 2.

The data obtained show that submarine groundwater is an important source of nutrient transfer to the Global Ocean. More than 3 kg of nitrate-nitrogen is supplied from the karst cavity per day, which can lead to eutrophication and the primary production being limited by phosphorus in the local sea site.

### 3.3. Vulnerability of Submarine Groundwater of Karst Origin under Anthropogenic Influence

The nitrate, phosphate, and nitrite concentrations calculated for the groundwaters are 5–7 times higher than for the pure spring water in this region, formed in the feeding zone (Figure 5). For comparison, the Ayazma-Chokrak spring was used, located in the mountains in a specially protected natural area near the site of submarine discharge; the data are given in Table 4. The concentrations in spring water are even higher than the concentrations of nutrients at some stations in seawater; in particular, in the innermost part of the karst cavity. According to [39], the springs are formed in the zones of aeration and seasonal level fluctuations, i.e., in the mountains in the region of study. These territories are less susceptible to anthropogenic impact. However, the submarine discharge is formed in the zone of full saturation. Taking into account the data of geological studies [40] on the transit of groundwater formed at the Ai-Petry Massif (Figure 5) through the Baydarskaya Valley (Skelsky spring), the anthropogenic factor causes high concentrations of nutrients in submarine groundwater. The villages of the Baydarskaya Valley with a population of more than 8000 people are not connected to sewerage networks, and agriculture and animal husbandry are developed, which can cause groundwater pollution [41,42].

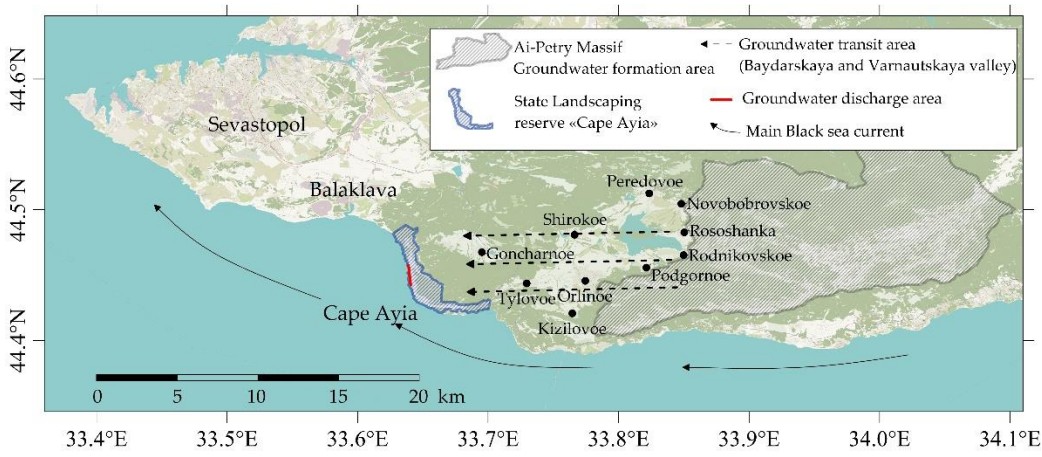

**Figure 5.** Areas of formation, transit, and discharge of groundwater in southwestern Crimea.

**Table 4.** The concentration of nutrients in conventionally clean underground waters of the Ayazma-Chokrak spring.

| Concentration, μM | | | | | |
|---|---|---|---|---|---|
| **Si** | **DIP** | **TDP** | **NH$_4^+$** | **NO$_2^-$** | **NO$_3^-$** |
| 142.8 | 0.06 | 0.06 | 1.25 | 0.18 | 6.34 |

The question about the pollution of karst groundwater as a result of human economic activity has been raised more than once by researchers [42]. The approach to its description was proposed by the scientists of the V.I. Vernadsky Crimean Federal University [43]. It is necessary to measure the concentration of other pollutants (pesticides, heavy metals) to clarify the scale of anthropogenic influence.

*3.4. Assessment of the Possibility of Capturing Submarine Springs of Cape Ayia*

This issue has at least three aspects—environmental, economic, and technological.

The environmental aspect includes the condition of minimizing damage to the environment since Cape Ayia is part of a specially protected natural area and is one of the unique regions of the Crimean Peninsula. It is also known that areas with freshwater outlets are spawning grounds for fish. Therefore, the capturing of submarine water springs should not cause damage to the area as a whole.

The economic and technological aspects are related. According to various estimates, the flux of the main spring at Cape Ayia is at least 1 million m$^3$ per year. Such an amount of freshwater could improve the situation with water supply in the Sevastopol region. However, one of the difficulties is that it is quite far from the networks and consumers, which requires stretching a pipeline either from Balaklava or from Laspi Bay. As an alternative to constructing electrical networks, it is possible to use wind generators or solar panels near the spring.

According to the results obtained, brackish water comes out of the karst cavity in the surface layer 1 m thick, so it is possible to partially close the karst cavity at the exit at a level of +0.5−−1.5 m, which will prevent the seawater from entering the cavity, but will allow fish to enter. In the innermost part, it is necessary to install another dam from the bottom to the level of +0.5 m. In this area, freshwater will accumulate, after displacing salty water. From this part, freshwater will be pumped out into a pipeline laid in the direction of Balaklava or Laspi Bay, for use as technical water or with further treatment for drinking water. Another issue that potentially complicates the development of submarine springs is the possibility of seawater intrusion into the karst before the spring comes out [44]. In this case, the possibility of obtaining freshwater remains questionable.

In any case, this issue requires further research by hydrogeology specialists and design engineers of offshore hydraulic structures.

*3.5. Seasonal Variability of the Distribution of Parameters, Fluxes of Submarine Spring, and Nutrients*

The expeditions carried out in 2019 (spring-autumn) and 2020 (winter-summer) showed significant variability in the distribution of hydrological, hydrochemical, and radiochemical parameters.

This phenomenon is quite complex, and the variability of the distribution in its parameters is determined by the following main factors:

- the flux of the submarine spring;
- meteorological conditions during the expedition;
- hydrological conditions in the area;
- seasonal nature of intensity of the biogeochemical processes in seawater.

In turn, the flux of submarine springs is determined by the amount of precipitation where the groundwater forms.

The expeditions took place on windless days so that the absence of wind and wave effects reduced the mixing of light fresh water and dense seawater. The salinity profile data (Figure 3a) show that groundwater spreads from the spring in a 0.5–1 m thick layer. The minimum salinity value was found in the innermost part of the karst cavity (11‰) because mixing occurs at the exit of groundwater from the rock.

Further, the layer of brackish water distributes in the adjacent water area and the currents carry it in the southeast direction. Paper [13] indicates that changes in the hydrological and hydrochemical characteristics of surface waters occur at a distance of 0.8–1.2 km from the submarine spring.

A comprehensive analysis of the results obtained (Figure 3) shows that the minimum values of the flux of the submarine spring were observed in September 2019, and the maximum in June 2020. Table 2 shows the results of calculating the fluxes of submarine groundwater for various parameters, the average values, and fluxes of nutrients.

To explain the results obtained, we analyzed the data on precipitation in the area where the groundwater for the submarine springs forms—the plateau of Mount Ai-Petri [39,43] (Figure 6). The transit time of groundwater from the area of precipitation to the area of discharge measured by indicator methods is about 30 days [13]. To determine the relationship between the amount of precipitation according to the data of the meteorological station on the Ai-Petri plateau [45] and the flux of the submarine spring, the amount of precipitation a month before the expedition was determined.

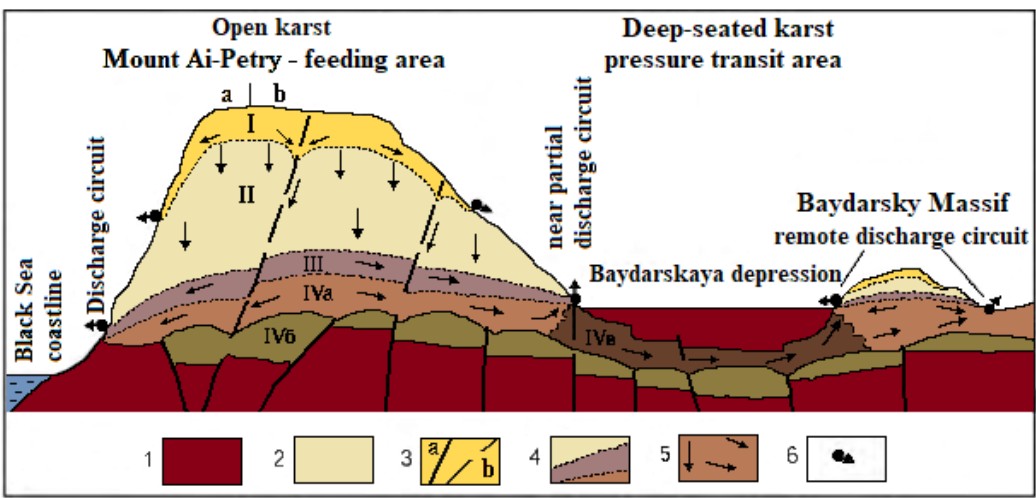

**Figure 6.** Formation of the groundwater flows at the southwest of the Crimean Peninsula. Adopted from [43]. Legend: 1—poorly permeable rocks, 2—karst rocks, 3—tectonic disturbances: 3a—in the basement, 3b—in karst rocks, 4—boundaries of hydrodynamic zones, 5—directions of groundwater motion, 6—karst springs. Karst massifs: a—basement coastal, b—slope continental. Hydrodynamic zones: I—epicarst (mostly scattered supply; non-pressure water forming a suspended horizon); II—aeration (vadose—mainly downward free water motion along the cracks and channels); III—seasonal level fluctuations (epiphreatic, alternating conditions of zones II and IVa); IV—full supply zone; subzones: IVa—mainly free-flowing waters of open karst with intense water exchange, with local pressure in the channels (phreatic); IVb—pressure water with slow water exchange; IVc—pressure waters (artesian pressure) of deep-seated karst with intense water exchange.

The amounts of precipitation according to the meteorological station on the Ai-Petri plateau for the month before the expedition were as follows:

- expedition 24 March 2019—68.8 mm (plus the income from melting snow cover from 59 to 9 mm);
- expedition 10 September 2019—18 mm;
- expedition 23 February 2020—335 mm;
- expedition 19 July 2020—169 mm.

The obtained values are generally consistent with the average values of submarine spring fluxes. The lowest flux values were determined in September 2019, the highest in February and July 2020.

The actual expeditionary results are discrete. Longer data series are needed to determine more precisely the relationship between rainfall and groundwater flows. This is planned to be carried out by us by setting up an autonomous buoy that measures salinity and current velocity in the apex of the grotto.

### 4. Conclusions

We used various hydrological (salinity), hydrochemical (silicon, nitrate-nitrogen), and radiotracer (radium concentration) methods to determine the flux of submarine spring discharge near Cape Ayia, in one of the powerful submarine springs at the southwestern coast of Crimea. Parameters such as the concentration of radium isotopes and nitrate ions, as well as the seasonal variability of the groundwater discharge fluxes and nutrients with them, were studied for the first time for this region. It was shown that, depending on the season, the flux ranges from 4.1 to 13.9 thousand $m^3$/day. The concentration of nutrients in the centers of discharge is much higher than in the background waters. This confirms that submarine water discharge is an important source of nutrient input into the sea. The discharge capacity qualitatively correlates with the amount of precipitation in the locations of formation of groundwater on the karst plateau of Mount Ai-Petri. The minimal fluxes of submarine groundwater were observed in autumn, since, taking into account the transit time of groundwater from the formation area, the previous month of August is traditionally the driest month.

**Author Contributions:** I.I.D.—participation in expeditionary work, sampling, the processing of samples for radium isotopes, discussion and analysis of the obtained results. O.N.K.—measurement of nutrient concentration, the processing of the obtained data, discussion and analysis of the obtained results. N.A.B.—participation in expeditionary work, sampling, discussion and analysis of the obtained results. I.G.S.—sorbent preparation, participation in expeditionary research, sampling. A.I.C.—participation in expeditionary work (hydrological survey), discussion and analysis of the results. I.G.T.—determination of the concentration of radium isotopes, discussion and analysis of the obtained results. All authors have read and agreed to the published version of the manuscript.

**Funding:** This work was carried out with the financial support of the state assignment of the Ministry of Education and Science of the Russian Federation (theme "Coastal research" No. 0555-2021-0005), and Sevastopol State University, project identifier 42-01-09/169/2021-7.

**Institutional Review Board Statement:** Not applicable.

**Informed Consent Statement:** Not applicable.

**Data Availability Statement:** Not applicable.

**Acknowledgments:** Not applicable.

**Conflicts of Interest:** The authors declare no conflict of interest.

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
