# Peer review of "Seasonal Variability of Nutrients and Radium Isotope Fluxes from Submarine Karstic Spring at the Southwest of Crimea, Black Sea"

_water, doi:10.3390/w14040568_

Round 1

Reviewer 1 Report

The paper is now linked to previous research. Proposed amendments are accepted.

Author Response

We are grateful to the reviewers for reading the manuscript and valuable comments aimed at improving it.

Reviewer #1

Comment.

The paper is now linked to previous research. Proposed amendments are accepted.

Answer. Thank you.

Reviewer 2 Report

  1. It is better not to use “;” in this sentence. Authors are suggested to reconstructed this whole sentence. You can divide it into two sentences if it is necessary.
  2. The second sentence is too long. Please make it shorter or divide it into two or more sentences.
  3. LN 16-18: This sentence is not needed for the abstract.
  4. LN 18-19: This sentence is poor to present the aims of your manuscript. It is too general and hard to see what is the special thing or novelty of the present work.
  5. The whole abstract should be reorganized. Authors use too many words to introduce why do this work, but lacks words for the results and findings of your research in present abstract. Please refer to reference “Investigating sources, driving forces and potential health risks of nitrate and fluoride in groundwater of a typical alluvial fan plain” to find how to organize the abstract.
  6. Keywords: “hydrophysical” is a keyword? Revise it.
  7. Keywords: It is better to use “radium isotope” to replace “226Ra, 228Ra”.
  8. The introduction is not like a normal academic paper’s one. Generally, you can introduce why this topic is needed to be studied or concerned in the first paragraph. And then, review the international progress of this topic worldwide, remember not only focus on your study area or around your study area as your present manuscript is for the international readers not only local readers. Thirdly, sometimes you can review tools for studying this topic. Fourthly, tell the aims of the present study. Authors can refer to reference “Investigating sources, driving forces and potential health risks of nitrate and fluoride in groundwater of a typical alluvial fan plain”, “Hydrogeochemical constraints on groundwater resource sustainable development in the arid Golmud alluvial fan plain on Tibetan plateau” to find how to organize the introduction section, and cite them and external ones to support your introduction.
  9. Ln 77-87 is recommended to be involved in the paragraphs after it.
  10. The study area section is lack. Geology and hydrogeology are also needed to be introduces.
  11. How about the accuracy of the samples? How about the QA/QC?
  12. LN 105-108: should be involved in the introduction section rather than here.
  13. Ln 110: the samples information should be involved in the sampling section.
  14. Where did you sample the water? Boreholes or caves? You need to introduce this in the sampling section.
  15. Methods like that in Ln 114-119 should be involved in the method section rather than the Results and Discussion section. Same to the other similar parts.
  16. This manuscript is like a report not an academic paper. Please reorganize the whole manuscript.
  17. Ln 317: the figure is not clear enough. There are too many explanations for legends in the figure name.
  18. For the conclusions section: authors are recommended to reconsider the novelty of your manuscript and what can you bring to the readers. The conclusions should involve the main results and findings of your research. The present one is not in good presentation.

Reviewer 3 Report

General: The text is in many places not clear. It is necessary to improve the English text. For instance the  authors may work at  Marine (not Maine) Hydrophysical Institute (Line 7).

Abstract: It is common that the authors make a general evaluation of the results of the work at the end of the abstract.

The aims of the work are usually explained at the end of Introduction. You describe that the water flux varied highly in submarine groundwater (line 64). Could you explain in lines 71-73 better what could it mean for practical or theoretical life? Why was this work as done? 

Line 81 given in? Or presented?

Line 83: Describe how salinity was determined and its possible determination error! In many parts you refer to salinity as S. However, S means in chemistry sulfur. Present therefore here salinity (S)! Now your abbreviation is misleading. 

You determined both radium isotopes which originate from natural sources and nutrients which are biogenic (as you correctly present). Could you claim that it least nitrogen and phosphorus compounds are anthropogenic since Sevastopol is not far and this area has rather a high population density and has been inhabited for a long time (line 79)?

In figure 3 a correct solinity to salinity! You must present what you mean with CSi and the other Cs. If it is only concentration, omit the C which is only misleading. Increase the font of texts in Fig. 3!

What is the idea of Tables 2 and 3?  If you aim to present that the flux varied highly at different seasons, turn both these Tables so that the sampling times form columns and the parameters form rows. In Table 3 the equations are the same as presented already in Fig. 3. It is not allowed to present the same data twice and thus, omit it from Table 3.

Lines 153-157 improve the text!

Fig 5: Improve the contrast! This journal allows color figures and now only the black rock is different from the others.

Line 171: Sewage water is informed the first time. I would inform this much earlier, maybe in Introduction, since it gives knowledge that there high anthropogenic load to this groundwater.  

Line 254: omit one sentence in the middle of this line.

Line 332: The precipitation of 18 mm is low if thinking about water collection in a climate where the temperature is rather high. If the climate will change, what is expected to be the precipitation in this area in the months with the lowest precipitations? Could you refer to this in your conclusions?

Reviewer 4 Report

The paper presents a lot of data on the hydrology and chemistry of a freshwater discharge in a cave in the Crimea peninsula. The authors calculate fluxes of elements (nutrients) to the sea and compare these values to literature data. So far so good.

The science is worth publishing, but the writing of the paper is so poor that it is difficult for the reader to understand what is going on. It requires rewriting, using a simpler style and accurate wording. As it is some parts are written in a journalistic style that does not fit in a scientific journal. Drastic editing is required.

As an illustration of the statement above, I have noted the following points (while many others could have been noted also):

  1. 71: replace outputting by "discharging in", and "fluxes of nutrients with SGD" by "fluxes of nutreints associated to the SGD".

Fig. 3: error: solinity should be salinity.

Table 2: what do you mean by a quantity of a chemical element expressed in cubic meters per day? Please define the unit. What is S? Sulfur?? What is the average? The average of what? See also line 151 "concentration of silicic acid, 1915 m3/day".

  1. 255-156: what do you mean? I see many more than 6 samples in Fig. 4. Do you mean 6 samples for each season?

Fig. 4 would be clearer if the Ra values were distinguished for each sampling period. May be there is a correlation with salinity at a given period, but changing with time.  IN ANY CASE DATA MUST BE MADE AVAILABLE TO READERS.

  1. 165 what is 106 in "RV Professor Vodyanitsky cruise"?? In this section the discussion would be clarified based on a figure representing the 228Ra/226Ra ratio as a function of salinity, instead of the 228Ra versus 226Ra as in Fig. 4c.
  2. 176: Please use adequate wording. Replace "The main pathway for the entry of silicic acid into groundwater is the leaching from rocks through which they pass" by " aqueous silica (or dissolved silicic acid) comes from the weathering of surrounding rocks", or something like that. Silicic acid does not "enter" the system. Same l. 195.
  3. 177. This true only for the cited case. Silica in natural waters is more variable than this.
  4. 181. Please use a simpler style. Simply write "The present data lead to the following relationships". Period.

Table 3. What is "the approximation"?

  1. 200: quite obscure sentence. Please reword.
  2. 205. Improper English. Rewrite.
  3. 208: "anthropogenic pollution". Do we have to understand that there is natural pollution??
  4. 22 Table 2. What is "the value of reliability the approximation" (awkward English, though)? Do you mean quality of the fit expressed as the standard deviation of fit?
  5. 232. Avoid journalistic style ("These values prove the opinion of some scientists that submarine groundwater makes a great contribution to the transport of nutrients at the land-ocean interface").
  6. 233: nitrate-nitrogen is not removed from the cave. Instead it should be written that the spring in the cave provides more than 5 kg of NO3 per day. Besides this one may ask "so what?".
  7. 236. Again awkward sentence. Why don't you write a simple sentence like "the nitrate phosphate and nitrite concentrations are 5-7 times higher for the groundwaters than for the spring water"? At this point what is the difference between groundwater and spring water?

L; 253. This is quite a general statement. It does not bring anything to the paper.

L 263. I do not understand. The area is proteted but the exploitation of the spring water in a spawning area should not cause any problem? I am confused.

  1. 317. Fig. 5. What do you call a closed karst? There are streamlines in the figure showing water circulation in this part of the graph.

Round 2

Reviewer 3 Report

Still a few things:

Line 14: omit from the start of the sentence "And besides" 

Line 19 add Salinity (S)  .. since S means usually sulfur.

Line 249 add nitrate, phospate ...  (comma)

Your figure 6 is now good! 

Reviewer 4 Report

My comments and requirements for changes have been properly addressed. The new manuscript has been quite improved, especially in its writing. I recommend publication.

Author Response

We are grateful to the reviewers for reading the manuscript and valuable comments aimed at improving it.

This manuscript is a resubmission of an earlier submission. The following is a list of the peer review reports and author responses from that submission.

Round 1

Reviewer 1 Report

Comments

The authors in section 2.2 say they have determined silicic acid, dissolved inorganic phosphorus, total dissolved phosphorus, ammonium, nitrites, nitrates but they don’t describe all in the section 2.6.

Line 85: specify storage and transport conditions

Line 110: indicate the references of the material used

Line 120: acronyms DIP e TDP have not been previously described

Line 120-124: Check and improve. The description is unclear.

Reviewer 2 Report

The paper “Submarine karstic spring as a source of nutrients and radium isotopes at the southwest of Crimea, Black Sea" is interesting, but nevertheless not acceptable for publication because, despite the changed title, the paper’s topic is to a large extent already contained in a paper published earlier under the title of “Studying Submarine Groundwater Discharge at the Cape Ayia: A Multi-Tracer Approach” (Dovhyi et al, 2021, in: Morskoy Gidrofizicheskiy Zhurnal 37/1) (http://physical-oceanography.ru/repository/2021/1/en_202101_04.pdf ), which hasn’t even been referenced in the proposed paper.   The new title implies a paper dealing with the distribution of nutrients and radium isotopes in part of the Black Sea. However, the paper analyses that aspect only to a minor extent: there is no connection with similar research and results either in the wider analysed area or worldwide, with the focus being precisely on the topic addressed in the earlier paper.   It is therefore suggested that the proposed paper be rejected. The authors are suggested to prepare a paper which will comply with the proposed title and to remove and only refer to the elements described in the earlier paper, and address in more detail the distribution of nutrients and radium isotopes in the coastal sea.